# Prolonged Sleep Deprivation Induces a Reprogramming of Circadian Rhythmicity with the Hepatic Metabolic Transcriptomic Profile

**DOI:** 10.3390/biology13070532

**Published:** 2024-07-17

**Authors:** Shiyan Liu, Kailin Zhuo, Yiming Wang, Xiaomei Wang, Yingying Zhao

**Affiliations:** Department of Physiology, School of Basic Medical Sciences, Health Science Center, Shenzhen University, Shenzhen 518060, China; 2100243061@email.szu.edu.cn (S.L.); 2021221036@email.szu.edu.cn (K.Z.); 13372889096@163.com (Y.W.); xmwang@szu.edu.cn (X.W.)

**Keywords:** sleep deprivation, circadian rhythm, locomotor activity, liver, metabolism

## Abstract

**Simple Summary:**

Sleep deprivation disrupts the natural sleep-wake cycle and can affect various physiological processes, including liver function and metabolism. In this study, we examined how five days of sleep deprivation in male mice altered their liver function and daily rhythms. We found that sleep deprivation led to changes in the mice’s sleep patterns and feeding behavior within the initial two days following sleep deprivation. Additionally, there was a noticeable decrease in 24-h serum glucose levels. Transcriptome analysis of the liver revealed significant alterations in the rhythmic expression of genes related to carbohydrate, lipid, and protein metabolism pathways. These findings indicate that prolonged sleep deprivation resets metabolic gene expression in the liver, potentially acting to support sustained wakefulness by ensuring adequate energy supply.

**Abstract:**

Sleep disturbances can disrupt the overall circadian rhythm. However, the impact of sleep deprivation on the circadian rhythm of the liver and its underlying mechanisms still requires further exploration. In this study, we subjected male mice to 5 days of sleep deprivation and performed liver transcriptome sequencing analysis at various time points within a 24-h period. Subsequently, we monitored the autonomic activity and food intake in these male mice for six days post-sleep deprivation. We observed alterations in sleep-wake and feeding rhythms in the first two days following sleep deprivation. Additionally, we also observed a decrease in 24-h serum-glucose levels. Liver transcriptome sequencing has shown that sleep deprivation induces the rhythmic transcription of a large number of genes, or alters the rhythmic properties of genes, which were then significantly enriched in the carbohydrate, lipid, and protein metabolism pathways. Our findings suggest that under conditions of prolonged sleep deprivation, the expression of metabolic-related genes in the liver was reset, leading to changes in the organism’s metabolic state to ensure energy supply to sustain prolonged wakefulness.

## 1. Introduction

The circadian clock effectively orchestrates behavior and physiology in response to environmental conditions. The endogenous circadian clock, which is primarily located in the suprachiasmatic nucleus (SCN) of the hypothalamus, orchestrates myriad physiological processes, including glucose metabolism, lipid homeostasis, and energy expenditure, to align with the 24-h day-night cycle. As a central regulator of energy metabolism, the circadian clock system controls the diurnal oscillation of genes that function in metabolic reactions in anticipation of, and preparation for, periodic and predictable changes [1,2,3,4].

Sleep is essential for health and has a profound impact on various metabolic pathways, particularly those related to glucose metabolism and insulin sensitivity [5,6,7,8,9]. During restorative sleep, the body optimizes glucose utilization and insulin action, thereby mitigating the risk of insulin resistance and type 2 diabetes [5]. Furthermore, sleep orchestrates the balance of critical appetite-regulating hormones, such as leptin and ghrelin, thereby modulating energy intake and expenditure. The sleep-wakefulness cycle is the most dominant feature of the body’s intrinsic 24-h rhythmicity [10]. Prolonged sleep deprivation can lead to a misalignment between the central circadian clock and peripheral metabolic pathways, resulting in metabolic dysregulation and an increased risk of metabolic disease [11,12,13,14,15]. Moreover, understanding the transition from acute to chronic sleep deprivation is crucial. While acute sleep loss has been shown to impair insulin sensitivity and disrupt appetite-regulating hormones [15,16,17], prolonged sleep deprivation exacerbates these effects, creating an obesogenic environment characterized by metabolic dysregulation and increased susceptibility to metabolic syndrome [18,19]. Therefore, scientific inquiry emphasizes the critical role of prioritizing sound sleep hygiene as a cornerstone in maintaining holistic metabolic health [15,20].

While it is known that sleep disorders can disrupt the overall circadian rhythms, relatively few studies have examined the impact of sleep deprivation on the circadian rhythms of the liver, a major site of nutrient and xenobiotic metabolism. Moreover, most sleep deprivation-related studies employ daily short-term sleep deprivation [21,22,23,24,25,26] (usually 6 h of sleep deprivation treatment during the day). In this study, we subjected male mice to prolonged sleep deprivation and comprehensively analyzed the effects of long-term continuous sleep deprivation on the metabolic rhythms of the mouse liver. Our findings not only elucidate the intricate interplay between sleep patterns and hepatic metabolic processes but also underscore the profound implications that these interactions hold in broader physiological and pathophysiological contexts.

## 2. Materials and Methods

### 2.1. Mice

Wild-type C57BL/6 male mice (aged 8–10 weeks) were purchased from Guangdong Medical Laboratory Animal Center (Guangzhou, China) (SPF, SCXK (Yue) 2018-0002, Guangzhou, China). The male mice were raised in a specific-pathogen-free animal facility under a 12:12-h light-dark cycle (lights on at 7 a.m. (defined as Zeitgeber Time zero (ZT0)) and lights off at 7 p.m.). The activities of male mice were not restricted, and they had free access to food and water. Working according to the random number table method, the male mice were divided into the following two groups: the Control group and the sleep deprivation (SD) group (n = 12 per group). The PER2-luciferase male mice that were used to evaluate peripheral PER2: LUC rhythms during a 24-h period were obtained from the Jackson Laboratory and were randomly divided into the Control and SD groups (n = 3 per group). The animal study was reviewed and approved by the Institutional Animal Care and Use Committee of Shenzhen University.

### 2.2. Sleep Deprivation

Mice in the SD group were subjected to continuous sleep deprivation for 5 days, using a sleep deprivation system (Pinnacle Technology, Lawrence, KS, USA) designed to gently restrict sleep without unnecessarily exercising the rodent. This is achieved by a rotating bar placed a short distance above the cage floor, which lightly nudges the animal awake and encourages low levels of activity until the animal maintains wakefulness on its own. It has been shown to be compatible with the classical sleep restriction paradigm [27,28] and can effectively deprive mice of sleep [28,29,30]. During sleep deprivation, the rod rotated every 30 s for a period of 30 s. The mice were confirmed to be awake every 4 h. If necessary, physical interference measures were applied to prevent mice from lying near the pole to avoid the metal rod. The Control group received no treatment. The activities of the mice were not restricted, and they had free access to food and water. After 5 days of sleep deprivation, serum and liver tissue samples were collected from the experimental and Control groups every 6 h within a 24-h period (i.e., at ZT0, ZT6, ZT12, and ZT18; n = 3 at each time point).

### 2.3. Blood Tests

Whole blood obtained from the mice was allowed to stand at room temperature for 1 h, followed by centrifugation at 5000 rpm for 15 min at 4 °C to extract the serum. The obtained supernatant was then diluted 5-fold with physiological saline and subjected to biochemical analysis of total protein (TP), albumin (ALB), total cholesterol (TC), total triglycerides (TG), aspartate transaminase (AST), and alanine aminotransferase (ALT) levels using a blood biochemistry analyzer (Mindray BS-220, Mindray, Shenzhen, China).

### 2.4. Intraperitoneal Glucose Tolerance Test (IPGTT)

After 5 days of sleep deprivation, we randomly selected 4 mice in the sleep deprivation group and the control group to fast for 12 h. At ZT12, we performed an IPGTT by injecting a 20% glucose solution dissolved in sterile saline at a dosage of 2 g of glucose per kg of body weight. After disinfection, tail-vein blood levels were measured at 0, 15, 30, 60, 90, and 120 min using a glucometer (ACCU-CHEK Performa, Roche, Indianapolis, IN, USA).

### 2.5. Locomotor Rhythm Recording

To record the behavioral rhythms of mice after 5 days of sleep deprivation, 3 mice were randomly selected from each of the SD and Control groups and placed into the behavioral activity observation box at the same time. The locomotor behavior and duration of eating of the mice were recorded and analyzed by using the animal video tracking software, Ethovision XT 14 (Nordes, Wageningen, The Netherlands), for 6 days with no restrictions to food and water access or activities throughout. Spontaneous motor activity was defined as the distance traveled per unit of time (3 min).

For the in vivo monitoring of the peripheral PER2: LUC rhythm during a 24-h period, the PER2: LUC mice in the Control group received no intervention, while mice in the SD group were deprived of sleep for 5 days. For the in vivo fluorescence analysis, each mouse received an intraperitoneal injection of d-fluorescein potassium salt in PBS at 150 mg/kg and prepared with fluorescein (concentration 15 mg/mL) [31]. After 2–3 min, the bioluminescence signal of the PER2 protein was measured every 4 h at ZT2, ZT6, ZT10, ZT14, ZT18, and ZT22, using a small animal fluorescence imaging system.

### 2.6. Total RNA Extraction

Total RNA was extracted from the liver tissue using the FastPure Cell/Tissue Total RNA Isolation Kit V2 (RC112, Vazyme, Nanjing, China). The concentration and purity of the total RNA were determined with an Epoch 2 microplate reader (BioTek, Winooski, VT, USA). Reverse transcription was conducted with the HiSlidTM cDNA Synthesis Kit for qPCR (with dsDNase) (Mikx, Shenzhen, China). The reverse transcription reaction system was as follows: 10× dsDNase buffer, 1 µL; total RNA, 1000 ng; and RNase-free ddH_2_O to make the volume of 10 µL. The reaction procedure was set at 37 °C for 2 min and at 65 °C for 2 min to remove genomic DNA contamination. Then, we added HiSlid^TM^ cDNA Synthesis MasterMix (4 µL) and RNase-free ddH_2_O to make up a volume of 20 µL. The reaction procedure was set at 50 °C for 15 min and at 85 °C for 5 min.

### 2.7. Real-Time PCR

The reverse transcription gDNA samples were used as templates for real-time PCR (RT-PCR) using the 2× Polarsignal color qPCR mix (Mikx, Shenzhen, China) in a TOptical gradient quantitative thermocycler (Biometra, Gottingen, Germany) sequence detector. The reaction system for PCR was as follows: 2× Polarsignal^®^ color qPCR mix, 10 µL; forward primer, 0.5 µL (10 µM); reverse primer, 0.5 µL (10 µM); template DNA, 2 µL; and ddH_2_O to make the volume of 20 µL. PCR reaction procedure was set at 94 °C for 20 s, 40 cycles of 94 °C for 10 s, and 60 °C for 20 s. The relative expression of the target gene was calculated using 2−ΔΔCt methods.

### 2.8. RNA-Seq

The transcriptome sequencing process of the mouse liver samples was performed according to the BGI DNBSEQ Eukaryotic Transcriptome resequencing project protocol. Total RNA was extracted from the tissues using TRIzol (Invitrogen, Carlsbad, CA, USA). The quantity and quality of total RNA were determined using the NanoDrop and Agilent 2100 bioanalyzer (Thermo Fisher Scientific, Waltham, MA, USA). The total RNA was enriched using the mRNA enrichment method to construct a DNA library; the library quality was then checked using the Agilent Technologies 2100 bioanalyzer. The final library was formatted after being qualified. The final amplified DNA nanosphere (DNB) was loaded into the pattern nanoarray with Phi29, and a peer-end 100-base reading was generated on the BGIseq500 platform (BGI-Shenzhen, China) [32]. The raw data were filtered with SOAPnuke (v1.6.5) [33] by: (1) removing reads containing adapters (adapter contamination); (2) removing reads where the unknown base (‘N’ base) ratio is more than 1%; (3) removing reads with a low-quality base ratio (base quality less than or equal to 15) that is more than 40%. Afterward, the clean reads were obtained and stored in FASTQ format.

### 2.9. Statistical Analysis

All analyses were performed using GraphPad Prism 8.0.2 (GraphPad Software, San Diego, CA, USA) and R [34]. All data were shown as the mean ± standard deviation. Group comparisons were performed with an unpaired Student’s *t*-test. *p* < 0.05 was considered to indicate statistical significance.

### 2.10. RNA-Seq Data Analysis

The clean data were mapped to the reference genome and assembled unique genes by Rsubread [35], and the expression level of the genes was calculated using the featureCounts function. Between-group differential gene analysis was performed using DESeq2 [36] under the conditions of an adjusted *p*-value of ≤0.05. The JTK_CYCLE algorithm [37] was used to identify the periodic characteristics of each group, as described previously [38]. The delta value and period are 6 and 24, respectively, which can detect periodic features within a 12–24-h period with an ‘independent’ method. Based on a 24-h oscillation period (BH.Q < 0.05), transcripts were divided into two categories: rhythmic genes (JTK algorithm, BH.Q < 0.05) and non-rhythmic genes (JTK algorithm, BH.Q ≥ 0.05). Other parameters were set to the default. The JTK_CYCLE algorithm determined the genes with rhythmic change, and the nonlinear cosine regression tool CircaCompare [39] was used to detect the amplitude (i.e., half the difference between the peak and trough of a given response variable), Mesor (the midline statistic for estimating rhythms), and acrophase (the time at which the response variable peaks) rhythm characteristics. In comparisons between the SD and Control groups, the rhythm was considered to have changed if at least one of the three rhythm characteristics was significantly different (*p* < 0.05) between the two groups. To elucidate the change in phenotype, GO enrichment analysis of annotated different expression genes was performed by clusterProfiler [40].

## 3. Results

### 3.1. Prolonged Sleep Deprivation Alters the Locomotor-Rest and Feeding Rhythms in Mice

To explore whether prolonged sleep deprivation affects activity and dietary rhythm, mice were randomly selected from the Control and SD groups after five consecutive days of sleep deprivation and observed in an autonomous activity recorder for six days (Figure 1A). We observed significant weight differences between the two groups of mice on day 5 of the sleep deprivation regimen and the first two days post-deprivation (Appendix A). This suggests that long-term sleep deprivation can reduce body weight in mice, which is consistent with previous findings [41]. We also observed significant changes in the overall locomotor activity of mice in the SD group, particularly in the initial two days post-sleep deprivation (Figure 1B,C). As nocturnal creatures, mice in the SD group exhibited a notably reduced proportion of night-time activity on the first day compared to the Control group. Conversely, the proportion of night-time activity of the SD group on the second day was markedly elevated relative to that of the Control group. By the third day, the locomotion–rest circadian rhythm of the mice began to stabilize (Figure 1C).

The changes in the cumulative average locomotor activity of the SD mice over a period of six days after the sleep deprivation regimen was applied were relatively subtle (Figure 1D–F). Surprisingly, sleep deprivation had a more marked effect on feeding rhythm. We observed that the feeding duration of SD mice was significantly lower than that of Control mice from ZT8 to ZT24. However, the peak of feeding duration still appeared in the first half of the dark phase (Figure 1G–I), although there were no significant differences in overall changes in food intake (Appendix A). Both feeding and locomotor activity rhythms showed similar oscillation patterns as those that were observed in the nocturnal animals.

### 3.2. Prolonged Sleep Deprivation Alters Metabolism and Liver Function in Mice

To assess the effects of prolonged sleep deprivation on metabolism and liver function, we examined the serum components of mice in the Control and SD groups after five days of sleep deprivation. The results revealed no significant differences in total protein (TP) and albumin (ALB) levels between the groups (Figure 2A–D), indicating no significant changes in hepatic protein synthesis function. Total cholesterol (TC) and triglyceride (TG) levels also did not differ significantly (Figure 2E–H), suggesting that there were no significant alterations in liver lipid metabolism function. However, the TC and TG levels in SD mice showed a decreasing trend, corresponding to variations in feeding times (Figure 2G–I). In the SD group, alanine aminotransferase (ALT) levels exhibited an upward trend but did not reach statistical significance (Figure 2I,J), whereas aspartate aminotransferase (AST) levels significantly increased overall and at multiple time points (Figure 2K,L), suggesting potential liver or organ damage. SD mice showed significantly decreased serum glucose levels at multiple time points within a 24-h period (Figure 2M,N) and maintained lower blood glucose levels during intraperitoneal glucose tolerance testing (IPGTT) (Figure 2O,P), indicating significantly enhanced liver glucose metabolism capacity in sleep-deprived mice.

### 3.3. Peripheral Per2 Rhythms Exhibited Subtle Changes Following Prolonged Sleep Deprivation

To understand the circadian rhythm of clock gene expression in the peripheral organs of mice, we injected luciferin (15 mg/kg body weight) into PER2: LUC mice for 8 min and performed non-invasive fluorescent imaging of the submandibular glands, liver, and kidneys. The Per2 protein maintained its 24-h rhythmicity in the liver, kidneys, and submandibular glands, with lower levels detected during the day and higher levels at night (Figure 3A–C), which is consistent with previous research [42,43]. We found that the level of PER2 protein in the liver of SD group mice showed an overall downward trend; however, there was no significant difference in the rhythmic variation of the Per2 protein between the two groups. Subsequently, we investigated whether there were alterations in the rhythmicity of the other clock genes in the hepatic transcriptome.

### 3.4. Prolonged Sleep Deprivation Alters the Liver Transcriptome Profiles of Mice across Different Time Points

To further investigate the specific mechanisms underlying the effects of prolonged sleep deprivation on mice, we compared the hepatic transcriptomes of the Control and SD groups at ZT0, 6, 12, and 18 (Figure 1A). Principal component analysis revealed the significant clustering of the two groups at all four time points (Figure 4A,E,I,M). By comparison of the two groups, we detected 470, 136, 51, and 7831 differentially expressed genes (DEGs) in the SD group at ZT0, 6, 12, and 18, respectively (Figure 4B,C,F,G,J,K,N,O). The largest number of differentially expressed genes between the two groups was observed at ZT18 (Figure 4N,O). Subsequently, we performed gene ontology (GO) enrichment analysis on the DEGs at each time point and found significant enrichment in processes related to energy molecule synthesis and metabolism. For example, at ZT0 and ZT6, there was significant enrichment seen in processes such as protein folding, carboxylic acid biosynthetic, and fatty acid metabolic processes (Figure 4D,H). At ZT18, there was significant enrichment in processes such as ribonucleoprotein complex biogenesis, the generation of precursor metabolites and energy, and autophagy (Figure 4P). Although there was enrichment in processes such as iron ion transport and immune response at ZT12 (Figure 4L), the specificity was not highly significant.

### 3.5. Prolonged Sleep Deprivation Induces an Increase in Rhythmic Gene Expression

To evaluate the effect of prolonged sleep deprivation on mouse liver transcriptome oscillations, we conducted a transcriptomic analysis of the liver tissues that were collected every 6 h for a 24-h period (ZT0, ZT6, ZT12, and ZT18). Principal component analysis revealed apparent intra-group clustering and significant inter-group separation (Figure 5A), making it suitable for downstream analysis. Subsequently, we used the JTK-cycle [37] algorithm to identify rhythmic gene sets in the Control and SD groups, and further utilized the CircaCompare [39] analysis algorithm to assess the intersection of these two gene sets. For genes within the intersection, if any of the three major rhythmic features (amplitude, MESOR, and acrophase) changed within the two groups, they were considered to be genes with altered shared rhythms. We observed the following: 1483 genes only exhibited rhythmic expression in the control group, defined as Part I; 2905 genes only exhibited rhythmic expression in the sleep-deprived group, defined as Part IV; 253 genes showed rhythmicity in both groups without changes, defined as Part II; and 1102 genes displayed rhythmicity in both groups with altered rhythms, defined as Part III (Figure 5B). Thus, prolonged sleep deprivation was shown to induce significant changes in the oscillation patterns of rhythmic genes (Figure 5C,E,G,I).

To explore the impact of rhythmic gene alterations on the physiological functions of the liver, we conducted GO analysis on the four groups of genes. Prolonged sleep deprivation led to a loss of rhythmicity in genes that are associated primarily with pathways such as RNA localization, the establishment of RNA localization, and nucleic acid transport (Figure 5D). This loss of rhythmicity may disrupt the normal regulation of gene expression, leading to impaired liver function. In contrast, pathways with maintained rhythmicity included nucleocytoplasmic transport, nuclear transport, and protein import into the nucleus (Figure 5F). Genes that exhibited changes in rhythmicity were predominantly found in pathways that are related to the regulation of protein catabolic processes, the regulation of ER stress, and small-molecule catabolic processes (Figure 5H). The dysregulation of these pathways may contribute to the development of liver diseases, such as fatty liver disease and liver fibrosis. Interestingly, genes that acquired rhythmicity due to prolonged sleep deprivation were concentrated in the liver metabolism pathways, such as energy metabolism, the proteasome-mediated ubiquitin-dependent protein catabolic pathway, autophagy, and fatty acid metabolism (Figure 5J). The acquisition of rhythmicity in these pathways may help the liver adapt to the stress of sleep deprivation by increasing energy production and protein turnover.

### 3.6. Prolonged Sleep Deprivation Resets the Rhythmicity of Multiple Metabolic Pathways

We subsequently selected representative pathways from the prolonged sleep deprivation-induced acquisition of new rhythms (III) and those genes with acquired rhythms (IV) for rhythmic analysis. The majority of genes in the ‘regulation of protein catabolic process’ pathway underwent a rhythm phase shift, with the trough moving from ZT12 to ZT18 (Figure 6A,B). Under normal conditions, the genes in the ‘insulin-mediated glucose metabolism’ pathway were divided into two groups: one peaked at ZT12 and the other had a trough at ZT12. However, after prolonged sleep deprivation, one group peaked at ZT18, while the other had a trough at ZT18 (Figure 6C,D). According to the JTK-cycle analysis, the ‘generation of precursor metabolites and energy’ and ‘fatty acid metabolic process’ pathways were determined to be arrhythmic under normal conditions. After prolonged sleep deprivation, most genes in these pathways exhibited a trough around ZT5 and another near ZT16 (Figure 6E,F). Interestingly, after prolonged sleep deprivation, the ‘fatty acid metabolic process’ acquired a similar rhythm, with a trough at ZT5 and another trough around ZT16 (Figure 6G,H). These findings provided evidence that prolonged sleep deprivation reset the rhythmicity of multiple metabolic pathways.

### 3.7. Prolonged Sleep Deprivation Synchronizes the Expression of a Large Number of Metabolic Genes

We compared the rhythmicity of individual genes under normal conditions and after prolonged sleep deprivation. Among the circadian clock-related genes, *Bmal1*, *Dbp*, *Npas2*, and *Rorc* maintained their rhythmicity, while *Clock*, *Cry2*, *Per2*, and *Rora* exhibited significant changes in their rhythmicity (Figure 7A,E). Genes involved in the ‘insulin-mediated glucose metabolism’ pathway, including *Akt2*, *Pten*, *Sirt1*, etc., showed significant changes in the amplitude and phase of their rhythm curves, with expression peaks or valleys near ZT5 and ZT16 (Figure 7B,F). Notably, genes within the ‘generation of precursor metabolites and energy’ and ‘fatty acid metabolic process’ pathways acquired rhythmicity after prolonged sleep deprivation, and most of them also reached peak or trough expression values near ZT5 and ZT16. (Figure 7C,D,F). Thus, our findings indicated that prolonged sleep deprivation synchronizes the expression of a large number of metabolic genes.

## 4. Discussion

Sleep plays a crucial role in metabolic regulation. To explore the effects of prolonged sleep deprivation on hepatic metabolic rhythms, we subjected mice to a 5-day sleep deprivation regimen and analyzed their liver transcriptomes using RNA-seq. Our analysis revealed that prolonged sleep deprivation disrupted the normal sleep-wake and feeding-fasting rhythms, leading to alterations in the rhythmic patterns of lipid and glucose metabolism. Notably, sleep deprivation effectively reset the rhythmic transcriptome of the mouse liver, resulting in a significant increase in the number of rhythmic genes expressed. This reset was accompanied by substantial changes in multiple metabolic pathways, including glucose metabolism, lipid metabolism, and autophagy. These findings highlight the remarkable plasticity of hepatic metabolic rhythms and their critical role in enabling an organism to adapt to sleep perturbations.

We hypothesize that the molecular mechanisms underlying these changes may involve alterations in the expression of key circadian clock genes, such as Per2 [44] and Cry2 [45], which regulate the rhythmic expression of metabolic genes. While the liver’s adaptive responses to long-term sleep deprivation demonstrate its resilience, the potential long-term consequences of these adaptations on overall health warrant further investigation.

Insufficient sleep and circadian misalignment can lead to an imbalance in energy metabolism [46,47,48,49]. Resting energy expenditure has been shown to vary with the circadian phase, reaching the lowest level in the late biology night period [50]. We observed that after 5 days of sleep deprivation, the mice exhibited signs of significant disruption to an activity-rest cycle. Furthermore, on the first day post-sleep deprivation, their daytime activity was notably higher than that of the Control group, especially during the light phase from ZT0–6.

Within the week following prolonged sleep deprivation, the mice displayed a significant increase in activity during the dark phase at ZT18–21, while their activity notably decreased during ZT21–24. Interestingly, despite no significant difference in daily food intake weight, the feeding duration of mice was significantly reduced across every time interval over 24 h. This may be attributed to the formation of habitual rapid food intake during the period of sleep deprivation. The timing of feeding behavior is governed by the central circadian clock located in the hypothalamus [51]. Under the regulation of this central timer, the peripheral liver clock integrates changes in murine feeding behavior to modulate hepatic metabolic transcription. In our study, we faced the challenge of disentangling the changes in the liver’s peripheral clock that are triggered by the central clock from those induced by food behaviors. However, the rhythmic changes that we observed in the liver metabolic transcriptome are orchestrated directly by the biological clock within each hepatocyte. In contrast to previous reports that sleep deprivation in human populations contributes to conditions such as overweight and obesity [52,53], we found that continuous sleep deprivation over 5 days in male mice reduced activity and feeding duration, and such a phenomenon has also been observed in previous studies concerning prolonged sleep deprivation in mice [41]. Monitoring the levels of hormones related to sleep and energy metabolism over a 24-h period will provide further crucial insights into the effects of sleep deprivation on energy metabolism.

Previous studies showed that sleep deprivation can lead to impaired glucose tolerance and reduced insulin sensitivity [5,6,7,8,9]. Short-term sleep deprivation usually leads to a decrease in the number of rhythmically expressed genes, such as those in the peripheral blood [20], lung [22], synapse [23], and liver [24]. A study in 2012 demonstrated that short-term sleep deprivation (6 h of sleep deprivation during the day) disrupted the rhythmicity of genes related to the lipid, carbohydrate, and protein metabolism pathways in the livers of mice [24]. In our study, prolonged sleep deprivation profoundly reshaped the rhythms of the mouse liver transcriptome. Prolonged sleep deprivation induced alterations in rhythmic patterns, leading to the emergence of 2950 genes with novel rhythmic expression. Intriguingly, these genes also predominantly participated in metabolic pathways, encompassing energy metabolism, proteasome-mediated protein degradation, autophagy, and fatty acid metabolism. Conversely, 1102 genes that originally exhibited rhythmic expression underwent changes in their rhythmicity post-sleep deprivation. Notably, these genes are primarily involved in protein regulation, insulin-mediated glucose metabolism, and steroid hormone-mediated signaling pathways. We observed that prolonged sleep deprivation led to an increase in the number of rhythmically expressed liver genes, which is an opposite result to that observed in short-term sleep deprivation studies in previous research; however, both studies have observed the significant enrichment of differentially expressed genes in the energy metabolism pathways related to lipids, proteins, and carbohydrates. These differences may be associated with factors such as the specific methods and timing of sleep deprivation. Further validation and discussions in future studies are necessary to determine the precise factors contributing to this discrepancy. The interplay between sleep disruption and metabolic regulation remains an intriguing area for investigation.

Under normal conditions, the genes involved in insulin-mediated glucose metabolism pathways exhibit rhythmicity. However, after 5 days of sleep deprivation, these genes displayed novel rhythmic expression patterns. Interestingly, the majority of these genes, *Akt2*, *Capn10*, *Pik3r2*, *Prkcz*, *Ptpn1*, *Slc2a8*, etc., peaked in expression at ZT5 and showed a transcriptional trough around ZT16. In contrast, a few genes, such as *Pten*, *Sirt1*, etc., peaked at ZT18 and showed their lowest expression around ZT6. In the case of the ‘generation of precursor metabolites and energy’ and ‘fatty acid metabolism’ processes, genes that initially lacked rhythmic expression acquired rhythmicity following prolonged sleep deprivation. Notably, unlike the insulin-mediated glucose metabolism genes, some ‘generation of precursor metabolites and energy’ and ‘fatty acid metabolism’ genes peaked at ZT5 and troughed at ZT16, including *AKt1*, *Bax*, *Cox17*, *Dgat2*, *Fabp5*, *Ndufa7*, *Nr1h3*, *Sirt6*, and *Slc25a33*. Thus, the mechanism underlying transcriptional regulation in rhythmicity in hepatic metabolism requires further investigation.

Although sleep deprivation has a significant impact on metabolic rhythm, our transcriptomic analysis revealed that prolonged sleep deprivation did not disrupt the rhythmic expression of critical core clock genes, such as *Bmal1*, *Dbp*, *Npas2*, and *Rorc*, while other genes such as *Clock*, *Cry2*, and *Per2* exhibited altered rhythmicity. Notably, prolonged sleep deprivation treatment decreased the expression of the PER2 protein in the liver, especially at night, but this did not constitute a statistical difference. In the analysis of transcriptional levels, we also observed that prolonged sleep deprivation treatment changed PER2 mRNA expression levels during the day and night, but only decreased during the day. This difference may be due to a combination of factors. On the one hand, bioluminescence analysis of male mice is a non-invasive experimental method that enables comprehensive monitoring of PER2 protein expression levels, but in the process, the analysis may be affected by environmental factors, such as individual differences in male mice, light, ambient temperature, etc. On the other hand, the process of transcription of per2 into translation in the body may be affected by other proteins in the body and may not be fully synchronized.

Circadian clocks have a fundamental role in maintaining liver homeostasis by regulating processes such as lipid and glucose metabolism, bile acid synthesis, and the expression of enzymes involved in drug metabolism and energy production. In this study, we present evidence highlighting the liver as a pivotal organ in the body’s adaptation to environmental changes. Our experimental results indicate that prolonged sleep deprivation induces significant alterations in the circadian rhythm of the mouse liver, with metabolic pathways related to glucose and lipid metabolism acquiring rhythmicity. The modulation of metabolic rhythms in the liver appears to be an adaptive response to long-term sleep alterations. The regulatory mechanisms that we have identified in this study provide a foundational understanding of clinical observations linking sleep disorders to metabolic diseases in the human population.

## 5. Conclusions

In conclusion, our study demonstrates that sleep deprivation significantly disrupts the circadian rhythm of the liver, leading to alterations in metabolic gene expression and changes in the organism’s metabolic state. The liver transcriptome sequencing analysis revealed that sleep deprivation induces the rhythmic transcription of a large number of genes, particularly those involved in the carbohydrate, lipid, and protein metabolism pathways. Furthermore, our results show that sleep deprivation resets the rhythmicity of multiple metabolic pathways, synchronizing the expression of a large number of metabolic genes to ensure an energy supply to sustain prolonged wakefulness. These findings provide new insights into the impact of sleep deprivation on the liver’s circadian rhythm and its underlying mechanisms, highlighting the importance of sleep in maintaining metabolic homeostasis. Further studies are needed to explore the long-term consequences of sleep deprivation on liver function and overall health, as well as the potential links between sleep deprivation, circadian rhythm disruptions, and the development of metabolic disorders.

## Figures and Tables

**Figure 1 biology-13-00532-f001:**
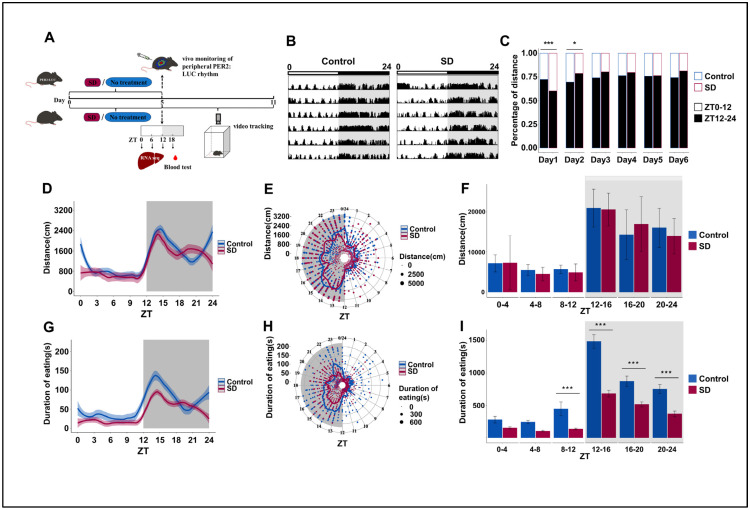
Prolonged sleep deprivation alters behavior rhythms and metabolism in mice. (**A**) Schematic diagram of the experimental procedure. (**B**) Representative actograms illustrating the locomotor activity rhythms of mice in the Control and sleep-deprived (SD) groups during a 24-h period after sleep deprivation, measured on each of 6 consecutive days (shown top to bottom = 24 h/line); the top bar represents the light-dark cycle. Locomotor activity is defined as the moving distance per unit of time (3 min). (**C**) Stacked bar charts showing the percentage of cumulative activity distance between the light-on and light-off phases for each of 6 consecutive days after sleep deprivation (n = 3 mice per group). (**D**–**F**) Activity distance recordings, as detected by the locomotor activity recorder over 6 days after sleep deprivation (n = 3 mice per group). (**G**–**I**) Cumulative feeding time recordings over 6 days after sleep deprivation (n = 3 mice per group). Data represent the mean ± standard deviation. Statistical significance represents the comparison between the Control and SD groups, as per an unpaired Student’s *t*-test. * *p* < 0.05, and *** *p* < 0.001 (see Appendix A).

**Figure 2 biology-13-00532-f002:**
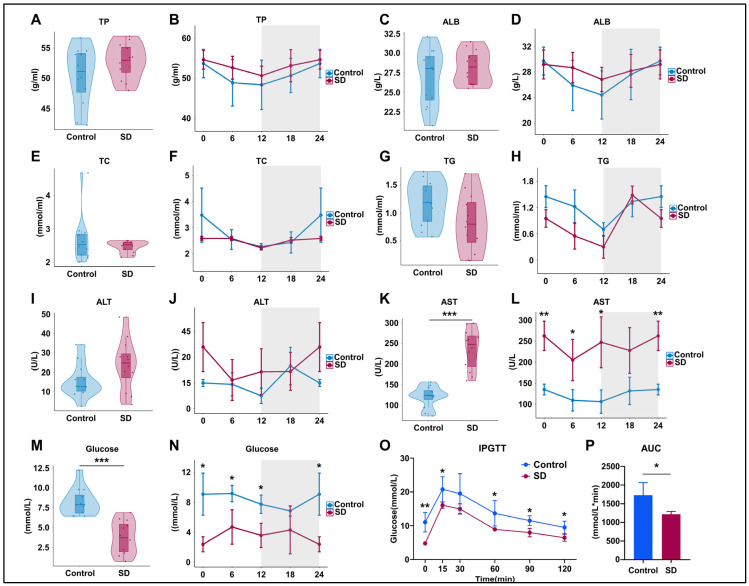
Prolonged sleep deprivation alters metabolism in mice. (**A**–**N**) Serum biochemical analysis of mice in the sleep-deprived (SD) and control (Control) groups (for violin plots, n = 12 per group of mice, and for line plots, n = 3 per group of mice at each time point); data for ZT24 in the line graph are replicates of ZT0. (**O**,**P**) Intraperitoneal glucose tolerance test (IPGTT) data (n = 4 mice per group). Total protein (TP), albumin (ALB), total cholesterol (TC), total triglycerides (TG), aspartate transaminase (AST), and alanine aminotransferase (ALT). The upper and lower ends of the violin plot are the maximum and minimum values of the data, respectively. Bar chart and line chart data represent the mean ± standard deviation. Statistical significance represents the comparison between the Control and SD groups, as per an unpaired Student’s *t*-test. * *p* < 0.05, ** *p* < 0.01, and *** *p* < 0.001 (see Appendix A).

**Figure 3 biology-13-00532-f003:**
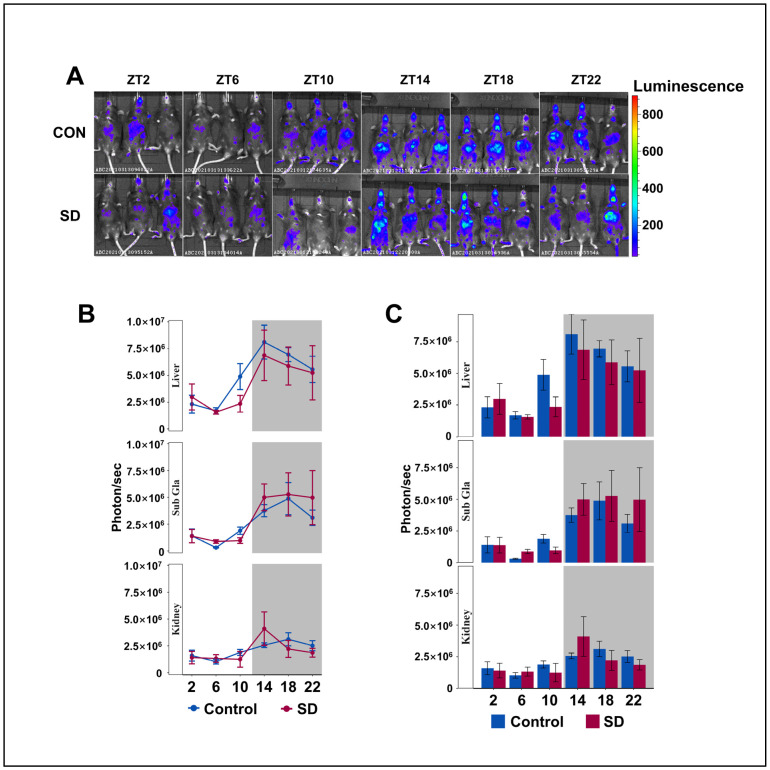
Peripheral Per2 rhythms exhibited subtle changes following prolonged sleep deprivation. (**A**) Fluorescent images of the submandibular gland, liver, and kidneys of PER2: LUC mice in the sleep-deprived (SD) and control (CON) groups at 2–3 min after an injection of fluorescein (n = 3 per group of mice at each time point). (**B**,**C**) Quantification of the submandibular gland, liver, and kidney bioluminescence data in (**A**), shown in photons per second (n = 3 per group of mice at each time point). Data represent the mean ± standard deviation. Statistical significance represents the comparison between the Control and SD groups, as per an unpaired Student’s *t*-test. (see Appendix A).

**Figure 4 biology-13-00532-f004:**
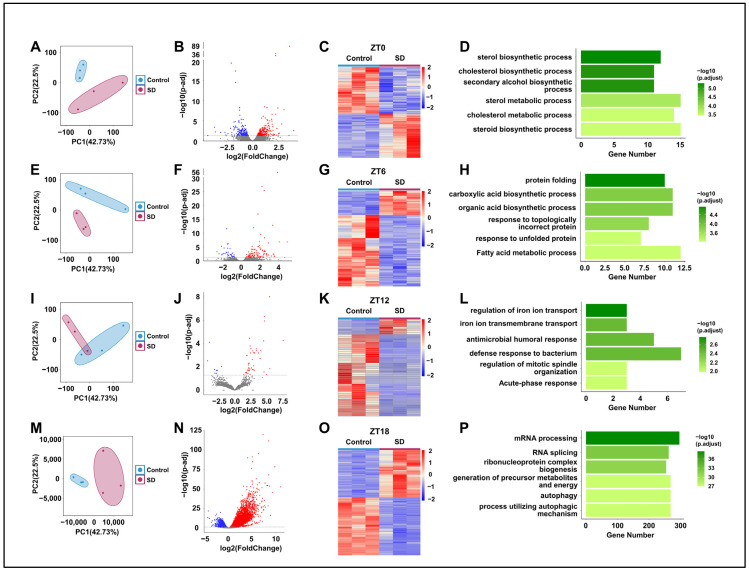
Prolonged sleep deprivation alters transcriptional gene expression at different times in the liver**.** (**A**,**E**,**I**,**M**) Principal component analysis of the differences in mouse gene expression data at ZT0, ZT6, ZT12, and ZT18 between the sleep-deprived (SD) and control (Control) groups. (**B**,**F**,**J**,**N**) Volcano maps comparing differentially expressed genes between the Control and SD groups and between the ZT0, ZT6, ZT12, and ZT18 data; upregulated and downregulated genes are shown as red and blue dots, gray dots indicate genes that are not significantly different, respectively (*p*-adjust < 0.05). (**C**,**G**,**K**,**O**) Heat maps showing the differentially expressed genes at ZT0, ZT6, ZT12, and ZT18 under Control and SD conditions. The expression level is expressed in the heat map as the Z-score. Each sample is represented by a column, and each gene is represented by a line. (**D**,**H**,**L**,**P**) Gene ontology functional annotation bar chart depicting the genes that show differential expression at the ZT0, ZT6, ZT12, and ZT18 time points under Control and SD conditions.

**Figure 5 biology-13-00532-f005:**
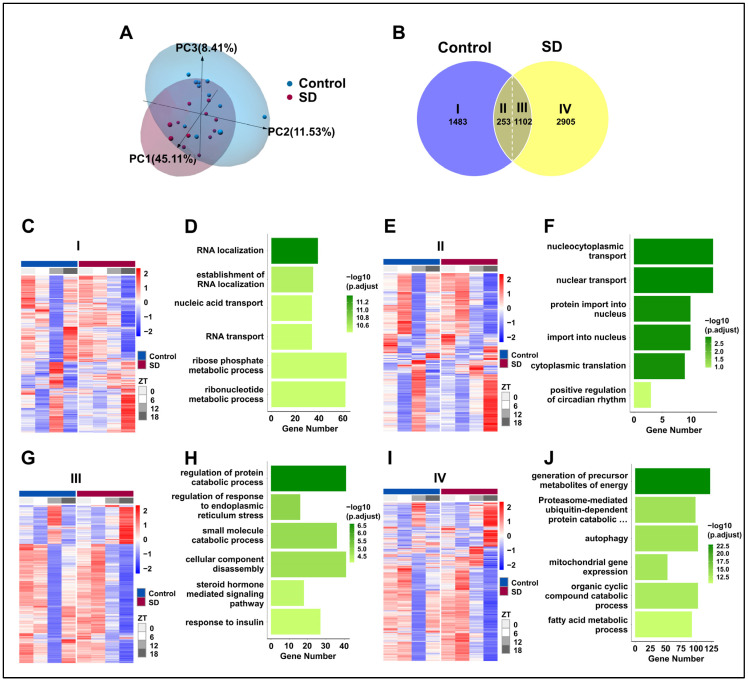
Prolonged sleep deprivation induces an increase in rhythmic gene expression. (**A**) Principal component analysis of samples in the sleep-deprived (SD) and control (Control) groups. (**B**) Venn diagram displaying the number (top) and ratio (bottom) of oscillating genes in liver samples isolated from mice in the SD and Control groups. (**C**–**J**) Gene ontology pathway enrichment and cluster analysis of the I, II, III and IV liver group genes. (**C**,**D**) Rhythmic transcripts unique to the Control group (I). (**E**,**F**) Rhythmic transcripts shared by the Control and SD groups, without significant changes (II). (**G**,**H**) Transcripts of rhythmic differences shared by Control and SD (III). (**I**,**J**) Rhythmic transcripts unique to the SD (IV).

**Figure 6 biology-13-00532-f006:**
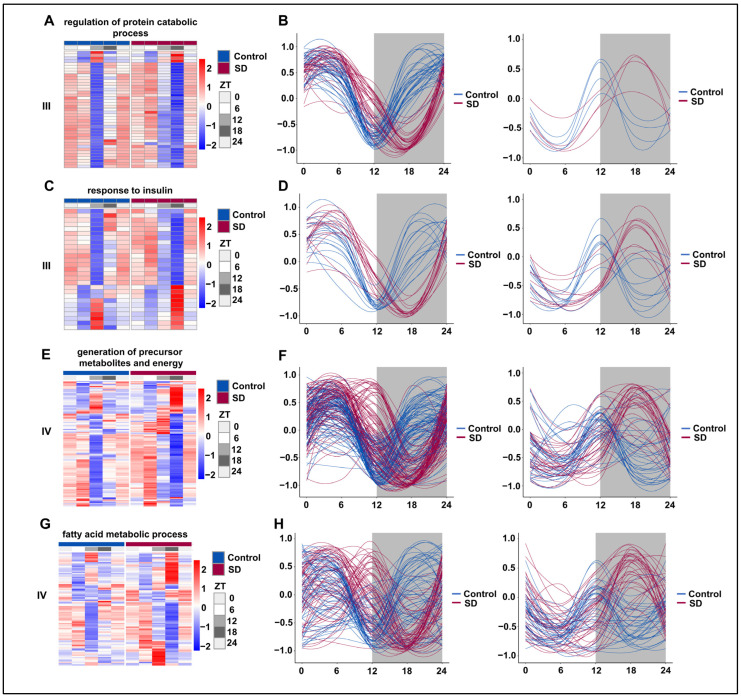
Prolonged sleep deprivation alters the circadian rhythm of multiple metabolic pathways. Heatmaps and fitted curve plots depicting the regulation of protein catabolic process genes (from III): (**A**,**B**) the response to insulin (from III); (**C**,**D**) the generation of precursor metabolites and energy (**E**,**F**) and fatty acid metabolic process genes (from IV); (**G**,**H**) heat map of genes enriched in the sleep-deprived (SD) and control (Control) groups. Each sample is represented by a column, and a line represents each gene. Each line in the fitted curve represents a gene; the expression of each gene is standardized to a range of −1 to 1, and the horizontal axis represents the ZT time. The data for ZT24 in the figure are the data for the replicated ZT0.

**Figure 7 biology-13-00532-f007:**
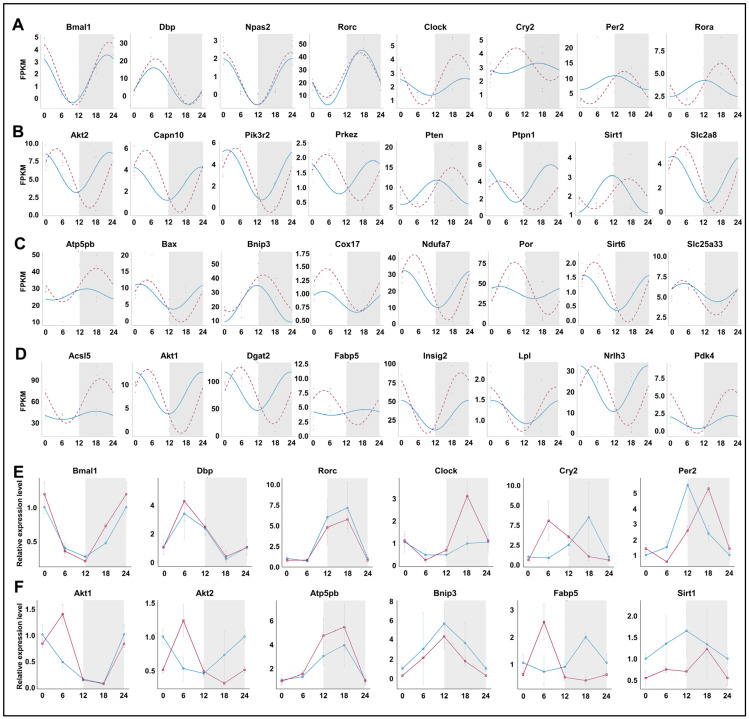
Prolonged sleep deprivation synchronizes the expression of a large number of metabolic genes. Diurnal expression of (**A**) core clock genes; (**B**) the response to insulin genes; (**C**) generation of precursor metabolites and energy genes; (**D**) lipid metabolism genes; qPCR results for the diurnal expression of core clock genes (**E**) and metabolic-related genes (**F**). The blue line indicates the Control group, and the red line indicates the sleep-deprived (SD) group. The horizontal axis represents ZT time. The grey background represents night.

## Data Availability

The total RNA-seq data are stored in the NCBI database with the project number ID PRJNA1020535. Any additional information required to reanalyze the data that are reported will be shared by the lead contact upon request.

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
