# Peer review of "Prolonged Sleep Deprivation Induces a Reprogramming of Circadian Rhythmicity with the Hepatic Metabolic Transcriptomic Profile"

_biology, 2024, doi:10.3390/biology13070532_

Round 1

Reviewer 1 Report

Comments and Suggestions for Authors

Dear Corresponding Author, thank you for submiting your paper and congratulations on your work. Below, i provide my review that adresses the Biology journal guidelines to give you a perspective that might lead to your integrative reflection.

Brief Summary:
This study thoroughly examines the effects of prolonged sleep deprivation on circadian rhythms and the metabolic transcriptional profile of the liver in mice. Using a multidisciplinary aproach combining behavioral, biochemical, and RNA sequencing analyses, the authors demonstrate that 5 days of sleep deprivation lead to significant alterations in activity rhythms, feeding patterns, and hepatic gene expression. Notably, the study reveals an increase in the number of genes with rhythmic expression in the liver after sleep deprivation, with significant changes in key metabolic pathways. These findings provide new and important insights into the mechanisms of metabolic adaptation in response to chronic sleep loss.

General Comments:
The manuscript presents original and well-conducted research that significantly contributes to our understanding of the interaction between sleep and metabolism. The strengths include the robust experimental design, the use of advanced techniques such as RNA-seq, and a thorough data analysis. The English language, although not native, seemed very accurate and perfectly explanatory to me.

However, there are some aspects that could be improved, which I report below for the authors' consideration:

  • The discussion could benefit from a section that more deeply explores the clinical implications of the results, particularly in relation to metabolic disorders associated with chronic sleep deprivation in humans, if possible to evaluate.
  • Greater contextualization of the results with respect to previous studies on short-term sleep deprivation would be useful. A comparative table might help highlight the key differences.
  • The statistical analysis is generally solid, but in some cases (e.g., Fig. 2) it might be useful to consider using two-way ANOVA to simultaneously evaluate the effects of treatment and time, or at least better specify the reason for the choice made.

Specific Comments:

  • Line 50-57: The transition between the effects of acute sleep deprivation and those of chronic deprivation could be smoother. Consider adding a linking sentence for better understanding.
  • Figure 1: Panels D-F and G-I could benefit from a slight reorganization to improve readability, the units of measurement on the axes are not clear. Also, it would be helpful to add clearer indications of light/dark phases directly in the graphs.
  • Line 219-234: This section on biochemical results is dense with information. While agreeing with the layout, try considering breaking it down into subsections to improve clarity.
  • Figure 3: The y-axis in graphs B and C is not clearly labeled. Add units of measurement and specify exactly what the values represent; the same anomaly is present in other graphs, try to see (if it's possible) to modify.
  • Line 308-331: The GO analysis is very interesting, but could benefit from a brief explanation of the biological significance of these changes in rhythmic gene expression.
  • Line 376-389: This part of the discussion on circadian rhythm alterations is also particularly interesting. Consider expanding it slightly, perhaps including hypotheses on the molecular mechanisms involved.
  • Line 414-437: The comparison with short-term sleep deprivation studies is crucial. A summary table highlighting the main differences in results could be useful but not determinant.

In conclusion, this is a high-quality manuscript that provides important new insights into the link between chronic sleep deprivation and hepatic metabolism.

I personally believe that the work is extremely worthy of publication, i only ask for some minor revisions, mainly focused on data presentation and discussion so that the impact and clarity of the work could be further improved.

Author Response

Comments 1: Fig. 2 might be useful to consider using two-way ANOVA to simultaneously evaluate the effects of treatment and time.

Response1: Thank you for your valuable comments on our research. Regarding the statistical analysis method you mentioned, we understand that ANOVA with repeated measures is a more robust and comprehensive approach, especially when dealing with inter-group comparisons at multiple time points. However, we chose to use unpaired T-tests to analyze the differences across time points based on the characteristics of our study design and questions and our analysis purposes.

Our research focuses on the independent effects of each time point, rather than just the overall time series change. We believe that using unpaired T-tests can better highlight the differences between these time points and help us understand the results more clearly, and some similar studies have chosen the same analytical method[1-3].

Comments 2: Line 50-57: The transition between the effects of acute sleep deprivation and those of chronic deprivation could be smoother. Consider adding a linking sentence for better understanding.

Response 2: Thank you for your careful review and valuable suggestions. We have revised lines 50-57 of the article to: “Moreover, understanding the transition from acute to chronic sleep deprivation is crucial. While acute sleep loss has been shown to impair insulin sensitivity and disrupt appetite-regulating hormones [15-17], prolonged sleep deprivation exacerbates these effects, creating an obesogenic environment characterized by metabolic dysregulation and increased susceptibility to metabolic syndrome [18,19]. Therefore, scientific inquiry emphasizes the critical role of prioritizing sound sleep hygiene as a cornerstone in maintaining holistic metabolic health [15,20]”

Comments 3: Figure 1: Panels D-F and G-I could benefit from a slight reorganization to improve readability, the units of measurement on the axes are not clear. Also, it would be helpful to add clearer indications of light/dark phases directly in the graphs.

Response 3: Thank you for your careful review and valuable suggestions. We have slightly readjusted the position of Figure 1, added labels to the X-axis, and included indicators for light and dark phases in Figures F and I.

Comments 4: Line 219-234: This section on biochemical results is dense with information. While agreeing with the layout, try considering breaking it down into subsections to improve clarity.

Response 4: Thank you for your modification suggestions, we have modified the corresponding content to:“To assess the effects of prolonged sleep deprivation on metabolism and liver func-tion, we examined the serum components of mice in the Control and SD groups after 5 days of sleep deprivation. The results revealed no significant differences in total protein (TP) and albumin (ALB) levels between groups (Figure 2A-D), indicating no significant changes in hepatic protein synthesis function. Total cholesterol (TC) and triglyceride (TG) levels also did not differ significantly (Figure 2E-H), suggesting no significant alterations in liver lipid metabolism function. However, TC and TG levels in SD mice showed a decreasing trend, corresponding to variations in feeding times (Figure 2G-I). In the SD group, alanine aminotransferase (ALT) levels exhibited an upward trend but did not reach statistical significance (Figure 2I, J), whereas aspartate aminotransferase (AST) levels significantly increased overall and at multiple time points (Figure 2K, L), sug-gesting potential liver or organ damage. SD mice showed significantly decreased serum glucose levels at multiple time points within 24 hours (Figure 2M, N) and maintained lower blood glucose levels during intraperitoneal glucose tolerance testing (IPGTT) (Figure 2O, P), indicating significantly enhanced liver glucose metabolism capacity in sleep-deprived mice.” See line 220-234 of the manuscript.

Comments 5: Figure 3: The y-axis in graphs B and C is not clearly labeled. Add units of measurement and specify exactly what the values represent; the same anomaly is present in other graphs, try to see (if it's possible) to modify.

Response 5: Thanks for your valuable feedback, we have modified the Y-axis label of Figure 3 (B, C) to Photons per second.

Comments 6: Line 308-331: The GO analysis is very interesting, but could benefit from a brief explanation of the biological significance of these changes in rhythmic gene expression.

Response 6: Thanks to your valuable revision suggestions, we have revised the results of the GO enrichment analysis in the manuscript as: “To explore the impact of rhythmic gene alterations on the physiological functions of the liver, we conducted GO analysis on the four groups of genes. Prolonged sleep deprivation led to a loss of rhythmicity in genes associated primarily with pathways such as RNA localization, establishment of RNA localization, and nucleic acid transport (Figure 5D). This loss of rhythmicity may disrupt the normal regulation of gene expression, leading to impaired liver function. In contrast, pathways with maintained rhythmicity included nucleocytoplasmic transport, nuclear transport, and protein import into the nucleus (Figure 5F). Genes that exhibited changes in rhythmicity were predominantly found in pathways related to the regulation of protein catabolic processes, regulation of ER stress, and small molecule catabolic processes (Figure 5H). The dysregulation of these pathways may contribute to the development of liver diseases, such as fatty liver disease and liver fibrosis. Interestingly, genes that acquired rhythmicity due to prolonged sleep deprivation were concentrated in liver metabolism pathways, such as energy metabolism, proteasome-mediated ubiquitin-dependent protein catabolic pathway, autophagy, and fatty acid metabolism (Figure 5J). The acquisition of rhythmicity in these pathways may help the liver adapt to the stress of sleep deprivation by increasing energy production and protein turnover.” See line 311-327 of the manuscript.

Comments 7: Line 376-389: This part of the discussion on circadian rhythm alterations is also particularly interesting. Consider expanding it slightly, perhaps including hypotheses on the molecular mechanisms involved.

Response 7: Thank you very much for your valuable suggestions to our research. We would be happy to expand our discussion further based on your suggestions. The following is a revised paragraph: “Sleep plays a crucial role in metabolic regulation. To explore the effects of prolonged sleep deprivation on hepatic metabolic rhythms, we subjected mice to a 5-day sleep deprivation regimen and analyzed their liver transcriptomes using RNA-seq. Our analysis revealed that prolonged sleep deprivation disrupted the normal sleep-wake and feeding-fasting rhythms, leading to alterations in the rhythmic patterns of lipid and glucose metabolism. Notably, sleep deprivation effectively reset the rhythmic transcriptome of the mouse liver, resulting in a significant increase in the number of rhythmic genes expressed. This reset was accompanied by substantial changes in multiple metabolic pathways, including glucose metabolism, lipid metabolism, and autophagy. These findings highlight the remarkable plasticity of hepatic metabolic rhythms and their critical role in enabling the organism to adapt to sleep perturbations.

We hypothesize that the molecular mechanisms underlying these changes may involve alterations in the expression of key circadian clock genes, such as Per2[44] and Cry2[45], which regulate the rhythmic expression of metabolic genes. While the liver's adaptive responses to long-term sleep deprivation demonstrate its resilience, the potential long-term consequences of these adaptations on overall health warrant further investigation.” See line 382-395 of the manuscript.

 Comments 8: Line 414-437: The comparison with short-term sleep deprivation studies is crucial. A summary table highlighting the main differences in results could be useful but not determinant.

Response 8: Thank you for your review and valuable comments on our research results. You mentioned that the comparison with short-term sleep deprivation studies is a crucial point, and we fully understand its importance. We have elaborated our observations in the discussion section of the paper (lines 417-446 of the manuscript) and compared and discussed them with short-term sleep deprivation studies.

Reference:

  1. Guan, D.; Xiong, Y.; Trinh, T.M.; Xiao, Y.; Hu, W.; Jiang, C.; Dierickx, P.; Jang, C.; Rabinowitz, J.D.; Lazar, M.A. The hepatocyte clock and feeding control chronophysiology of multiple liver cell types. Science 2020, 369, 1388-1394, doi:10.1126/science.aba8984.
  2. Kuang, Z.; Wang, Y.; Li, Y.; Ye, C.; Ruhn, K.A.; Behrendt, C.L.; Olson, E.N.; Hooper, L.V. The intestinal microbiota programs diurnal rhythms in host metabolism through histone deacetylase 3. Science 2019, 365, 1428-1434, doi:10.1126/science.aaw3134.
  3. He, W.; Tang, H.; Li, Y.; Wang, M.; Li, Y.; Chen, J.; Gao, S.; Han, Z. Overexpression of Let-7a mitigates diploidization in mouse androgenetic haploid embryonic stem cells. iScience 2024, 27, 109769, doi:10.1016/j.isci.2024.109769.

Reviewer 2 Report

Comments and Suggestions for Authors

This manuscript is well-written and reports on the effects of sleep deprivation on the circadian rhythm of the liver. The authors demonstrated alterations in metabolic gene expression after 5 days of sleep deprivation, which appear to reflect a mechanism aimed at supporting sustained wakefulness by ensuring an adequate energy supply. The topic is interesting due to the scarcity of data on the effects of prolonged sleep deprivation on the circadian rhythm of the liver. Below are some comments:

Page 2; line 83: what was the rationale of choosing 5 days of sleep deprivation and is there an upper limit of sleep deprivation compatible with life in the studied mice species.

Page 2; line 90: How was confirmation made that animals were awake?

Page 3; line 103-108: the number of mice that underwent the IPGTT and their origin were not specified

Page 6; line 233: How were lower glucose levels in the SD group compared to control group (as illustrated in figure 1O-P) interpreted as abnormal liver glucose tolerance?

Page 6; line 237: according to the methods section, the total number of mice per group is 12, not 16.

Page 6; line 238: according to the methods section, n=3 per group at each time point, not 4 mice.

Page 6; line 239: regarding the IGPPT, the number of mice was not specified in the methods section.

Page 9; line 308:  Please, refer to “Figure 5 B”.

Author Response

Comments 1: Page 2; line 83: what was the rationale of choosing 5 days of sleep deprivation and is there an upper limit of sleep deprivation compatible with life in the studied mice species.

Response 1: Thank you for your question, and in response to your question about why we chose 5 days of sleep deprivation, our response is as follows: People with severe sleep disorders[1] and those requiring sustained military operations[2,3] often face the challenge of being unable to sleep for many consecutive days. Studies have been conducted to assess cognitive function[4], motor capacity[3], and immune function[5] in these populations. In the current mouse sleep deprivation model, there are many studies on intermittent sleep deprivation, using more than 5 days[6-8], 14 days[9] or even longer[10], but there are few studies on continuous sleep deprivation for several days. Therefore, we choose to conduct five-day sleep deprivation on mice.

Comments 2: Page 2; line 90: How was confirmation made that animals were awake?

Response 2: Thanks for your question, for sleep deprivation experiments in mice, we subjected mice to sleep deprivation using a sleep deprivation instrument (Pinnacle Technology, USA) that has been shown to be compatible with the classical sleep restriction paradigm (see figure below). Moreover, several studies have shown that this method can significantly deprive mice of sleep[11-13], and we also check whether mice every four hours awake. We also line 90 mentioned in the manuscript.

Comments 3: Page 3; line 103-108: the number of mice that underwent the IPGTT and their origin were not specified

Response 3: Thanks for the questions you pointed out, we have explained in detail in lines 104-105 of the manuscript. We randomly selected 4 mice in the sleep deprivation group and the control group for IPGTT experiment.

Comments 4: Page 6; line 233: How were lower glucose levels in the SD group compared to control group (as illustrated in figure 1O-P) interpreted as abnormal liver glucose tolerance?

Response 4: Thank you for your question and we would like to elaborate further on this issue. Compared with the IPGTT results of mice in the control group, the blood glucose value and AUC value of mice in the SD group decreased significantly within 2h, indicating that glucose metabolism capacity increased significantly. We also revised the corresponding content in the paper as follows: “The results of IPGTT (Abdominal glucose Tolerance test) also showed significantly increased glucose metabolism in sleep-deprived mice”

Comments 5: Page 6; line 237: according to the methods section, the total number of mice per group is 12, not 16.

Response 5: Thank you for pointing out our mistake, we have revised the corresponding number of mice to 12.

Comments 6: Page 6; line 238: according to the methods section, n=3 per group at each time point, not 4 mice.

Response 6: Thank you for pointing out our problem with our manuscript. We are sorry that this is our input error. We have revised the number of mice tested at each time point to 3.

Comments 7: Page 6; line 239: regarding the IGPPT, the number of mice was not specified in the methods section

Response 7: Thank you for pointing out our problem with our manuscript, which we have detailed in lines 104-105 of the manuscript: 4 mice in the sleep deprivation group and the control group were randomly selected for IPGTT experiment.

Comments 8: Page 9; line 308:  Please, refer to “Figure 5 B”.

Response 8: Thanks for your revision suggestion, we have added "(Figure 5B)" in line 308 of the manuscript.

Reference:

  1. Cai, Y.; Chen, M.; Zhai, W.; Wang, C. Interaction between trouble sleeping and depression on hypertension in the NHANES 2005-2018. BMC Public Health 2022, 22, 481, doi:10.1186/s12889-022-12942-2.
  2. Gould, K.S.; Hirvonen, K.; Koefoed, V.F.; Røed, B.K.; Sallinen, M.; Holm, A.; Bridger, R.S.; Moen, B.E. Effects of 60 hours of total sleep deprivation on two methods of high-speed ship navigation. Ergonomics 2009, 52, 1469-1486, doi:10.1080/00140130903272611.
  3. Vaara, J.P.; Oksanen, H.; Kyröläinen, H.; Virmavirta, M.; Koski, H.; Finni, T. 60-Hour Sleep Deprivation Affects Submaximal but Not Maximal Physical Performance. Front Physiol 2018, 9, 1437, doi:10.3389/fphys.2018.01437.
  4. Boardman, J.M.; Bei, B.; Mellor, A.; Anderson, C.; Sletten, T.L.; Drummond, S.P.A. The ability to self-monitor cognitive performance during 60 h total sleep deprivation and following 2 nights recovery sleep. J Sleep Res 2018, 27, e12633, doi:10.1111/jsr.12633.
  5. Meier-Ewert, H.K.; Ridker, P.M.; Rifai, N.; Regan, M.M.; Price, N.J.; Dinges, D.F.; Mullington, J.M. Effect of sleep loss on C-reactive protein, an inflammatory marker of cardiovascular risk. J Am Coll Cardiol 2004, 43, 678-683, doi:10.1016/j.jacc.2003.07.050.
  6. Sutton, B.C.; Opp, M.R. Sleep fragmentation exacerbates mechanical hypersensitivity and alters subsequent sleep-wake behavior in a mouse model of musculoskeletal sensitization. Sleep 2014, 37, 515-524, doi:10.5665/sleep.3488.
  7. Barclay, J.L.; Husse, J.; Bode, B.; Naujokat, N.; Meyer-Kovac, J.; Schmid, S.M.; Lehnert, H.; Oster, H. Circadian desynchrony promotes metabolic disruption in a mouse model of shiftwork. PLoS One 2012, 7, e37150, doi:10.1371/journal.pone.0037150.
  8. Zhang, Y.M.; Wei, R.M.; Zhang, J.Y.; Liu, S.; Zhang, K.X.; Kong, X.Y.; Ge, Y.J.; Li, X.Y.; Chen, G.H. Resveratrol prevents cognitive deficits induced by sleep deprivation via modulating sirtuin 1 associated pathways in the hippocampus. J Biochem Mol Toxicol 2024, 38, e23698, doi:10.1002/jbt.23698.
  9. Baud, M.O.; Magistretti, P.J.; Petit, J.M. Sustained sleep fragmentation induces sleep homeostasis in mice. Sleep 2015, 38, 567-579, doi:10.5665/sleep.4572.
  10. Liu, X.; Yu, H.; Wang, Y.; Li, S.; Cheng, C.; Al-Nusaif, M.; Le, W. Altered Motor Performance, Sleep EEG, and Parkinson's Disease Pathology Induced by Chronic Sleep Deprivation in Lrrk2(G2019S) Mice. Neurosci Bull 2022, 38, 1170-1182, doi:10.1007/s12264-022-00881-2.
  11. Naylor, E.; Harmon, H.; Gabbert, S.; Johnson, D. Automated sleep deprivation: simulated gentle handling using a yoked control. Sleep 2010, 12, 5-12.
  12. Ahnaou, A.; Raeymaekers, L.; Steckler, T.; Drinkenbrug, W.H. Relevance of the metabotropic glutamate receptor (mGluR5) in the regulation of NREM-REM sleep cycle and homeostasis: evidence from mGluR5 (-/-) mice. Behav Brain Res 2015, 282, 218-226, doi:10.1016/j.bbr.2015.01.009.
  13. Naylor, E.; Aillon, D.V.; Barrett, B.S.; Wilson, G.S.; Johnson, D.A.; Johnson, D.A.; Harmon, H.P.; Gabbert, S.; Petillo, P.A. Lactate as a biomarker for sleep. Sleep 2012, 35, 1209-1222, doi:10.5665/sleep.2072.